# Identification of LncRNAs Involved in the Salt Stress Responses of *Eurotium cristatum* and Functional Analysis of Their Roles in Morphological Differentiation and Metabolic Regulation

**DOI:** 10.3390/biology14111592

**Published:** 2025-11-14

**Authors:** Yihan Wang, Zhenggang Xu, Meng Dong, Xiangdong Qing, Zhanjun Liu, Qinglin Zhang, Zhiyuan Hu

**Affiliations:** 1School of Materials and Chemical Engineering, Hunan City University, Yiyang 413000, China; wangyihan870428@163.com (Y.W.);; 2College of Forestry, Northwest A&F University, Yangling 712100, China

**Keywords:** *Eurotium cristatum*, lncRNA, salt stress, metabolic regulation

## Abstract

High salt stress impairs the growth and metabolite production of industrial microorganisms, including the tea-fermenting fungus *Eurotium cristatum*. This study explored the role of long non-coding RNAs (lncRNAs) in the salt stress response of *E. cristatum*. We identified 203 lncRNAs, 120 of which were differentially expressed under salt stress conditions. These stress-responsive lncRNAs are linked to key phenotypic changes, such as changes in the morphology and biosynthesis of valuable metabolites such as polysaccharides and melanin. Further analysis indicated that specific lncRNAs may act as molecular switches, modulating critical pathways in carbohydrate metabolism and stress responses. Our findings provide novel insights into fungal adaptation mechanisms and highlight potential genetic targets for engineering robust industrial strains with enhanced productivity, offering promising applications in the food and biotechnology industries.

## 1. Introduction

Long non-coding RNAs (lncRNAs) are a class of RNA molecules longer than 200 nucleotides that lack significant open reading frames (ORFs) and typically do not encode proteins [1]. Long non-coding RNAs were once dismissed as mere transcriptional “noise” or “junk DNA” [2]. However, advancements in high-throughput sequencing and bioinformatic tools have uncovered their role as key regulators in essential eukaryotic processes, such as differentiation, cell cycle, and metabolism [3]. Furthermore, the dysregulation of lncRNAs is linked to numerous diseases. Although most lncRNAs exhibit lower sequence conservation than their protein-coding counterparts [4], their expression patterns show strong spatiotemporal specificity and are highly responsive to environmental conditions [5], suggesting a potential role in the fine-tuned regulation of developmental signals and responses to environmental stimuli.

Regulatory mechanisms of lncRNAs are complex and diverse, primarily including the following [6,7]: (1) Chromatin modification and epigenetic regulation (recruiting modification complexes to influence histone marks or DNA methylation), (2) transcriptional regulation (acting as co-factors to interfere with or enhance transcription factor activity), (3) post-transcriptional regulation [binding to messenger RNAs (mRNAs) to affect their stability, splicing, or translation and functioning as competing endogenous RNAs to sequester microRNAs (miRNAs), thereby suppressing miRNA-mediated repression of target genes], and (4) protein function regulation (serving as scaffold molecules to mediate the formation of protein complexes or alter protein localization). Based on their mode of action, these mechanisms can be classified as cis-acting (regulating genes near their own genomic loci) or trans-acting (regulating distant genes through diffusion) [8].

LncRNAs play diverse roles in various processes, including growth and development, stress responses, and secondary metabolite synthesis in animals and plants [9,10,11]. However, compared to animal and plant lncRNAs, research on fungal lncRNAs remains limited. Their functions and mechanisms remain unclear, with current research predominantly focusing on model fungi such as *Saccharomyces cerevisiae* [12], *Aspergillus nidulans* [13], and some pathogenic fungi [14,15]. Moreover, systematic identification and functional exploration of lncRNAs in filamentous fungi of significant economic and ecological importance, particularly those involved in specialized food fermentation, such as *Eurotium cristatum*, are currently lacking.

*Eurotium cristatum*, commonly referred to as the “Jinhua fungus”, plays a crucial role in determining the quality of fermented teas, such as the Fuzhuan brick tea [16,17]. This microorganism undergoes two distinct morphological stages during its life cycle: the conidial (asexual) and ascospore (sexual) stages [18]. As a homothallic fungus, *E. cristatum* can self-fertilize and complete its sexual cycle independently. The shift between these stages is influenced by environmental factors, with osmotic pressure being a key regulator. High osmotic conditions suppress sexual reproduction and favor the asexual phase, while standard or hypotonic media encourage the development of sexual morphs [19,20]. This physiological response guided our experimental approach. We cultivated the sexual morph (G01) in a modified medium with reduced salt concentration and induced an asexual morph (G02) under high-salt stress. It is crucial to recognize that the phenotypic and transcriptomic differences analyzed in this study reflect an integrated response to environmental conditions. Specifically, the “salt stress response” of asexual morphs is inherently linked to salt-induced developmental reprogramming away from the sexual phase. Consequently, the observed differences likely encompass both direct osmotic adaptation and the effects of the morphological shift itself. Osmotic pressure influences the physiological state and metabolic profile of *E. cristatum*. For instance, the yield of the secondary metabolite lovastatin can be up to 20 times higher in the sexual form than in the asexual form [21], and extracts of sexual morphs exhibit significantly stronger antimicrobial activity than those of asexual morphs [22], indicating substantial differences in their metabolic pathways. However, the key molecular switches linking external environmental signals to the internal regulatory networks governing morphological transitions and metabolism, particularly the potential regulatory roles of lncRNAs, remain unclear.

The present study focused on the conidial and ascospore morphs of *E. cristatum*. Strand-specific lncRNA sequencing was performed to systematically identify lncRNAs in these two distinct physiological states and to analyze their basic sequence characteristics. Association analyses were conducted by integrating physiological indicators, such as mycelial biomass, melanin content, and polysaccharide content, to identify key candidate lncRNAs involved in regulating morphological differentiation and melanin and polysaccharide biosynthesis in *E. cristatum*. By investigating the potential roles of lncRNAs in salt stress responses and the regulation of morphological transition and secondary metabolism in *E. cristatum*, this study enhances our understanding of the post-transcriptional regulatory networks underlying various biological activities in filamentous fungi and provides a theoretical foundation and genetic resource for microbial strain improvement and directed regulation of fermentation using advanced molecular techniques.

## 2. Materials and Methods

### 2.1. Strain Culture and Sample Preparation

The *E. cristatum* strain (JH1805) utilized in this study was sourced from Hunan City University. Initially, the strain was activated on potato dextrose agar and then inoculated into a liquid medium with an inoculation amount of 1% under aseptic conditions for cultivation. The methods for cultivating the sexual (designated G01) and asexual (designated G02) morphs of *E. cristatum* are detailed below. The sexual morph (G01) was cultured statically (without shaking) in a modified potato dextrose liquid medium (300 g/L potato, 100 g/L glucose, and 0.5 mol/L NaCl) at 28 °C. Similarly, the asexual morph (G02) was cultured statically (without shaking) in a high-salt potato dextrose liquid medium [23] (300 g/L potato, 100 g/L glucose, and 3 mol/L NaCl) at 28 °C. For each morph, three independent biological replicates were cultivated under identical conditions. After 168 h of cultivation, mycelia were collected using sterile filter meshes, and their fresh weight was recorded. The mycelia were then thoroughly rinsed with sterile physiological saline, immediately frozen in liquid nitrogen, and subsequently stored at −80 °C.

### 2.2. Scanning Electron Microscopy

Samples of both sexual and asexual morphs of *E. cristatum* were prepared for SEM observation through the following process. Initially, the mycelial pellets were fixed in 2.5% glutaraldehyde overnight at 4 °C and then rinsed with a phosphate buffer (0.1 M, pH 7.0). The samples underwent progressive dehydration using a graded ethanol series (30%, 50%, 70%, 80%, 90%, 95%, and 100% × 3), with each step lasting 15 min. To preserve the delicate fungal structures, the dehydrated samples were subjected to critical point drying using liquid CO_2_. Subsequently, the dried samples were mounted on aluminum stubs, and their surfaces were sputter-coated with gold using an ion sputter coater to enhance conductivity [24]. Finally, the specimens were observed using a scanning electron microscope (SEM; HITACHI S-3000N; Hitachi High-Tech Corp., Tokyo, Japan). The dimensions of the fungal structures were measured using the ImageJ software (v1.53) [25].

### 2.3. Total RNA Extraction, Quality Control, and Library Construction

Strand-specific RNA sequencing was conducted on individual sexual (G01) and asexual (G02) mycelial samples (without biological replicates) to facilitate initial transcriptome assembly and lncRNA identification. Additionally, the expression patterns of key differentially expressed lncRNAs (DE-lncRNAs) were validated using reverse transcription polymerase chain reaction (RT-qPCR) (see Section 2.8).

Total RNA was isolated from 500 mg of mycelia (fresh weight) per sample using TRIzol reagent (Thermo Fisher Scientific, Waltham, MA, USA) according to the manufacturer’s instructions.

RNA Quality Control: The concentration and purity of RNA were determined using a NanoDrop ND-1000 spectrophotometer (Thermo Fisher Scientific, Waltham, MA, USA). RNA integrity was measured using an Agilent 2100 Bioanalyzer (Agilent Technologies, Santa Clara, CA, USA).

rRNA depletion and library construction: The Ribo-Zero rRNA Removal Kit (Illumina, San Diego, CA, USA) was used to eliminate ribosomal RNA (rRNA) from the total RNA sample. Strand-specific transcriptome sequencing libraries were fabricated using the NEBNext Ultra Directional RNA Library Prep Kit (New England Biolabs, Ipswich, MA, USA). The NEBNext Ultra Directional RNA Library Prep Kit employs the dUTP second-strand marking method to preserve the strand orientation information. Briefly, during the first-strand cDNA synthesis, random hexamer primers were used. Subsequently, during the second-strand synthesis, dTTP was replaced with dUTP. Following adapter ligation and polymerase chain reaction (PCR) amplification, the uracil-containing second strand was selectively degraded using the enzyme Uracil-Specific Excision Reagent (USER). This process ensures that only the first strand (which is complementary to the original RNA template) is amplified and sequenced, thereby allowing unambiguous determination of the transcriptional strand of origin for each resulting read. Library construction involved several steps. First, the RNA was fragmented. Complementary DNA (cDNA) was then synthesized. End repair was subsequently performed, followed by dA-tailing. Adapter ligation was the next step, which was succeeded by size selection. Finally, the libraries were enriched using PCR.

Library quality control and sequencing was performed using an Agilent 2100 Bioanalyzer to evaluate the quality of the constructed libraries, and the effective concentration was precisely measured using qPCR. Subsequently, libraries that met these requirements were sequenced on an Illumina HiSeq 2500 platform (Illumina, San Diego, CA, USA) to obtain PE150 paired-end reads.

### 2.4. Sequencing Data Quality Control and Alignment

The raw sequencing data underwent thorough quality control (QC). Initial QC reports were generated using FastQC (v0.11.9). Subsequently, adapter sequences, reads with an excessive number of ambiguous bases (N’s), and low-quality reads were filtered out using fastp (v0.23.4) [26]. MultiQC (v1.14) was used to aggregate and summarize the QC results from FastQC and Fastp into a single report. The resulting high-quality clean data were aligned to the *E. cristatum* reference genome (National Center for Biotechnology Information accession: ASM171748v1) using HISAT2 (v2.0.4) [27]. The strand-specific library parameter “--rna-strandness RF” was applied during alignment to improve the transcript strand orientation accuracy [28]. Alignment metrics, including overall and unique alignment rates, were calculated to ensure data suitability for subsequent analyses.

### 2.5. Transcript Assembly, LncRNA Identification, and Quantification

De novo transcript assembly for each sample was performed on BAM files generated using StringTie (v1.3.1) [29]. The transcripts of all the samples were merged using StringTie’s merge mode to create a unified set. Transcripts were compared with the reference genome annotation file using GFFcompare (v0.9.8) for transcript classification [30].

Next, lncRNA transcripts were identified by first filtering transcripts based on size (200+ nt), having at least two exons, and fragments per kilobase per million mapped reads (FPKM) ≥ 0.1. Transcripts with coding potential were assessed and filtered using CPC2, CNCI, CPAT, and Pfam scans [31]. This yielded the final set. StringTie was used to estimate the standardized expression levels of all genes and transcripts.

### 2.6. Differential Expression Analysis

Differential expression in *E. cristatum* samples was analyzed using edgeR (v 4.0) [32]. Genes or lncRNAs were regarded as differentially expressed if they met the criteria of an absolute value of log_2_(fold change) equal to or greater than one and a false discovery rate (FDR) of less than 0.01.

### 2.7. LncRNA Target Gene Prediction and Functional Enrichment Analysis

Cis-Acting Target Prediction: The Perl script was used to predict possible cis-acting target genes of the lncRNAs. This script pinpoints protein-coding genes situated within a 100-kilobase span either upstream or downstream of the genomic locus of the lncRNA [33].

Prediction of Trans-Acting Targets: The LncTarD (v 2.0) tool was used to predict the potential trans-acting target genes. This prediction was based on the free energy associated with base complementary pairing between lncRNAs and mRNAs. A normalized free energy threshold of less than −0.1 was applied for these predictions [33].

Functional Annotation and Enrichment Analysis: The Groups of mRNAs showing differential expression and the anticipated target genes of DE-lncRNAs were examined as described below. Gene Ontology (GO) enrichment analysis was performed using clusterProfiler 4.0 with a hypergeometric test to identify significant enrichment [34]. Kyoto Encyclopedia of Genes and Genomes (KEGG) pathway enrichment analysis was performed via KOBAS (version 3.0) employing a hypergeometric test, to pinpoint notable enrichment within metabolic and signaling pathways [35]. A significance threshold of FDR < 0.05 was set for all enrichment results.

### 2.8. RT-qPCR Validation of DE-LncRNAs

RT-qPCR with three independent biological replicates was employed to validate DE-lncRNAs detected via high-throughput sequencing. TRIzol reagent (Invitrogen, Carlsbad, CA, USA) was used to extract RNA from mycelial samples G01 and G02. The PrimeScript RT Reagent Kit with gDNA Eraser (Takara, Shiga, Japan) was used to eliminate genomic DNA and generate first-strand cDNA. Ten randomly selected differentially expressed lncRNAs were identified (Table 1). Primer Premier 5.0 software [36] was used to design specific primers. The relative expression level of DE-lncRNAs was calculated using the 2^−ΔΔCT^ method [37]. The expression of each target lncRNA was normalized to the stable internal reference gene β-Actin, and the sexual morph sample (G01) was designated as the calibrator. Therefore, the calculated relative expression represents the fold change in the asexual morph (G02) relative to G01.

### 2.9. Evaluation of Physiological Indicators in the Fermentation Broth

Three physiological indicators (mycelial growth rate, melanin content, and polysaccharide content) were quantitatively analyzed in the *E. cristatum* fermentation broth after seven days of cultivation.

Mycelial growth rate was assessed by measuring mycelial dry weight [38]. Briefly, after fermentation, the mycelia were collected, washed with distilled water, dried at 60 °C to a constant weight, and precisely weighed. The mycelial dry weight per unit volume of culture broth (g/mL) was then calculated.

Extracellular melanin in the fermentation broth was extracted using an alkaline extraction method [39]. Briefly, 50 mL of the fermentation broth was mixed with 70 mL of 1 mol/L NaOH solution and stirred in a 74 °C water bath for extraction. The mixture was then filtered and heated. The filtrate was acidified to pH 2.5 using 6 mol/L HCl, and allowed to stand in a 74 °C water bath for 4 h to precipitate melanin. The precipitate was collected by centrifugation at 8000× *g* for 15 min at 4 °C, washed to neutralize the pH with distilled water, and freeze-dried to obtain crude melanin. Crude melanin was purified via sequential extraction using ethyl acetate, chloroform, and ethanol. Purified melanin was freeze-dried and weighed, and the melanin content per unit volume of culture broth was calculated.

Polysaccharides in the fermentation broth were extracted using water extraction and alcohol precipitation [40]. Briefly, 50 mL of the fermentation broth was filtered to remove mycelia and insoluble impurities. Activated carbon was then added to the filtrate to adsorb the pigments. The decolorized filtrate was concentrated using a rotary evaporator at 55 °C. Proteins were removed from the concentrate by shaking with Sevag reagent at 160 rpm for 30 min, followed by centrifugation. Polysaccharides were precipitated from the remaining liquid by adding anhydrous ethanol and incubation at 4 °C for 24 h. The precipitate was collected via centrifugation at 8000 rpm, thoroughly dried in a vacuum dryer at 40 °C, and precisely weighed. Finally, the polysaccharide content per unit volume of culture broth was calculated.

### 2.10. Statistical Analysis

The experimental data were analyzed using the SPSS 23 software [41]. Standard deviations were used to determine the sample variability. Differences were considered statistically significant at *p* < 0.05.

## 3. Results

### 3.1. Cultivation of E. cristatum Mycelia

The liquid fermentation characteristics of the two morphs of *E. cristatum* are shown in Figure 1. The sexual morph pellicle appeared golden yellow with a rough and dense surface, and dark brown oil-like exudates were occasionally observed in the central region (Figure 1A). In contrast, the asexual morph pellicle exhibited a relatively uniform gray-green color with a smoother, tighter, and finer surface than the sexual morph. Moreover, no exudates were produced, indicating a high degree of distinction between sexual morphs (Figure 1B).

Scanning electron microscopy revealed the sexual and asexual reproductive structures of *E. cristatum*.

Sexual reproductive structures: Cleistothecia exhibited a rough surface with numerous grooves, appearing spherical or ellipsoidal with a diameter of 80–160 µm (Figure 1C). Ascospores measured 3.5–5 µm × 4.5–6 µm, with distinct surface ornamentation characterized by rough sharp warts and two prominent “coronary” projections at the equatorial region of the spore (Figure 1D).

Asexual reproductive structures: Conidial heads were broom-like, with a diameter of 30–55 µm; each conidial chain consisted of 2–4 conidia (Figure 1E). Mature conidia were mostly ellipsoidal, with the outer wall covered in small spinose protrusions, measuring 3.2–3.6 µm × 4.4–5.0 µm (Figure 1F).

### 3.2. Sequencing Data Quality and Alignment Statistics

Strand-specific lncRNA sequencing of sexual (G01) and asexual (G02) mycelial samples of *E. cristatum* was performed. A total of 34.40 Gb of clean data were obtained. The data volume for each sample exceeded 17 Gb, and the percentage of Q30 bases was >93.71%, indicating high-quality sequencing data. Clean reads were aligned to the *A. cristatus* GZAAS20.1005 (PRJNA271918) reference genome. The alignment rates for G01 and G02 were 97.74% and 96.99%, respectively, demonstrating high data utility and suitability for subsequent analyses.

### 3.3. LncRNA Identification and Classification

Through comprehensive screening using the StringTie assembly and four potential coding prediction tools (CPC2, CNCI, CPAT, and Pfam), 203 high-confidence lncRNAs were identified. Based on their genomic locations relative to known genes, these lncRNAs were classified into four categories. Long intergenic non-coding RNAs (lincRNAs) were the most abundant lncRNAs (70.9%), followed by antisense (26.6%) and sense (2.5%) lncRNAs. No intronic lncRNAs were detected.

A comparison of the sequenced lncRNAs and mRNAs revealed that the lncRNA length was primarily between 400 and 1200 nt (Figure 2A), whereas the mRNA length was mainly between 400 and 2000 nt and above 3000 nt (Figure 2B). The predicted ORF lengths of the lncRNAs were mainly between 50 and 150 nt (Figure 2C), whereas those of mRNAs were predominantly between 200 and 400 nt (Figure 2D). Compared with mRNAs, lncRNAs of *E. cristatum* exhibited shorter transcript lengths, fewer exons, and shorter ORFs, which is consistent with the typical structural characteristics of lncRNAs. The classification of the 203 lncRNAs is summarized in Appendix A. Their nucleotide sequences are provided in Appendix A, and the corresponding genomic annotations are available in Appendix A.

### 3.4. DE-LncRNA Analysis

Using thresholds of |log_2_(fold change)| ≥ 1 and FDR < 0.01, 120 DE-lncRNAs were identified (Full details of the 120 DE-lncRNAs are provided in Appendix A). Among these, 57 were upregulated and 63 were downregulated in G02 (asexual morphs) compared to those in G01 (sexual morphs). The clustering heatmap of the DE-lncRNAs in Figure 3 clearly shows their expression patterns, indicating significant differences in lncRNA expression profiles under different salt stress conditions.

### 3.5. Target Gene Prediction and Functional Enrichment Analysis

Predictions were made for cis-acting target genes situated within a 100-kilobase range either upstream or downstream of the DE-lncRNAs. GO enrichment analysis was carried out, and the results indicated that these target genes were notably enriched in several biological processes. These included oxidation-reduction reactions (GO: 0055114), transmembrane transport (GO: 0055085), and carbohydrate metabolic pathways (GO: 0005975). KEGG pathway analysis (Figure 4) indicated that these target genes were primarily involved in many pathways, including carbon metabolism (ko01200), urine metabolism (ko00230), and peroxisome metabolism (ko04146).

Trans-acting target genes based on complementary base pairing were predicted using LncTar software. GO enrichment analysis showed that these genes were significantly involved in processes, such as DNA integration (GO: 0015074) and protein ubiquitination (GO: 0071947 and 2000060). KEGG analysis revealed enrichment in various pathways, including ribosome and ribosome biogenesis, in eukaryotes (Figure 5).

### 3.6. RT-qPCR Validation

To assess the accuracy of the RNA sequencing data, ten randomly selected DE-lncRNAs with significant differences in expression levels were validated using RT-qPCR. The results are shown in Figure 6. Gene expression levels determined via Illumina sequencing were largely consistent with those measured via RT-qPCR, as the same genes analyzed via both methods exhibited similar upregulation and downregulation trends. These results confirm that our transcriptome sequencing data were accurate and reliable for further analyses.

### 3.7. Association Analysis Between LncRNAs and Phenotypes

Physiological indicator measurements revealed significant differences in the growth and metabolite accumulation of *E. cristatum* under different culture conditions (Figure 7). Compared to the asexual mycelia (G02) fermented in the high-salt potato glucose liquid medium, sexual mycelia (G01) fermented in the standard potato glucose liquid medium exhibited superior physiological phenotypes, with 219.7% higher mycelial biomass (6.17 vs. 1.93 g/L, G01 vs. G02), 135% higher extracellular melanin content (3.22 vs. 1.37 g/L, G01 vs. G02), and 53.3% higher extracellular polysaccharide content (0.23 vs. 0.15 g/L, G01 vs. G02).

To elucidate the molecular regulatory mechanisms underlying these significant phenotypic differences, we analyzed the DE-lncRNAs and found that their expression patterns were highly associated with the phenotypic traits of *E. cristatum*. In G01, the lncRNA MSTRG.3124.1 (an antisense lncRNA) was abundant (FPKM = 281.48). However, its expression was significantly lower in G01 than in G02 (log_2_FC = −4.94), making it one of the most highly expressed lncRNAs that was downregulated in sexual morphs compared to asexual morphs.

In G02 cells, the three lncRNAs showed high expression levels. Among them, MSTRG.10627.3 (a lincRNA) exhibited the most significant expression difference (FDR = 2.55 × 10^−106^; log_2_FC = 10.53), with its expression level in G02 (FPKM = 25.57) far exceeding that in G01 (FPKM ≈ 0). MSTRG.4077.4 (a lincRNA lncRNA) also showed high expression in G02 cells (FPKM = 57.59) and significant differential expression (FDR = 2.95 × 10^−76^; log_2_FC = 4.78). Furthermore, MSTRG.4764.1 (a lincRNA lncRNA) maintained a relatively high expression in G02 (FPKM = 65.61), showing significant differential expression (FDR = 2.94 × 10^−41^; log_2_FC = 3.14). The expression patterns of these DE-lncRNAs showed a high correlation with phenotypic indicators such as mycelial biomass, polysaccharide yield, and melanin yield, suggesting their important regulatory roles in these physiological processes.

## 4. Discussion

In the present study, we integrated transcriptomic and phenotypic analyses to investigate the roles of lncRNAs in *E. cristatum* under two contrasting physiological conditions: a sexual morph cultivated under standard conditions and an asexual morph induced by high-salinity stress. We emphasize that these states result from a complex interplay, in which salt stress serves as an environmental signal that triggers developmental transitions. Consequently, the differential expression of lncRNAs and the associated metabolic profiles reported here reflect a combined response involving direct osmotic stress adaptation and extensive reprogramming inherent to alternative developmental programs (asexual vs. sexual). Our results suggested that lncRNAs function as key regulatory molecules in this integrated response network.

Salt stress significantly altered the lncRNA expression profiles of *E. cristatum*, resulting in the identification of 120 DE lncRNAs. This finding suggests the broad involvement of lncRNAs in the transcriptional regulatory responses of *E. cristatum* to adversity, which is consistent with previous reports of abiotic stress inducing extensive lncRNA expression in other fungal species. LncRNAs and associated transcription factors regulate tolerance to high salt (5.13 mol/L NaCl) in *Aspergillus sydowii* [42]. Notably, antisense lncRNAs accounted for a substantial proportion (26.6%) of DE-lncRNAs in *E. cristatum* in this study. This class of lncRNAs typically forms complementary pairs with coding genes on their sense strands, enabling the precise regulation of target gene expression at the transcriptional or post-transcriptional levels [43]. This mechanism may form the molecular basis for the rapid and precise adjustment of gene expression in *E. cristatum* under stressful conditions.

Functional enrichment analysis of cis-acting target genes associated with DE-lncRNAs offers valuable insights into phenotypic disparities. These target genes are significantly enriched in various biological processes. For instance, they were notably involved in the “oxidation–reduction process” (GO: 0055114), “transmembrane transport” (GO: 0055085), and “carbohydrate metabolic process” (GO: 0005975). High-salt environments cause cellular osmotic imbalances and reactive oxygen species bursts; therefore, the regulation of redox homeostasis and ion transport is crucial for cellular adaptation to stress. The expression patterns of related lncRNAs/target genes in the G01 strain were more conducive to maintaining intracellular homeostasis, thereby explaining its growth advantage, as evidenced by its high mycelial biomass (6.17 g/L). More importantly, enrichment of target genes in pathways, such as the “carbon metabolism” (ko01200) and “purine metabolism” (ko00230) pathways, strongly correlated with high polysaccharide yield (0.23 g/L) in the G01 strain. We hypothesized that highly expressed lncRNAs (e.g., MSTRG.3124.1) in G01 promote polysaccharide biosynthesis by regulating adjacent sugar metabolism-related genes under normal conditions.

To further investigate the association between the DE-lncRNAs, mycelial morphological differentiation, and metabolite synthesis phenotypes, several key lncRNAs were screened. These molecules were categorized into two groups: those specifically highly expressed in G01 (high-yield sexual mycelia), which may positively regulate metabolite synthesis, and those specifically highly expressed in G02 (low-yield asexual mycelia), which may negatively regulate related pathways. In interpreting these results, we noted that the regulatory significance of an lncRNA was not solely dictated by its abundance. In living organisms, lncRNAs with low copy numbers can exert critical and precise regulatory effects [44,45]. In this study, we prioritized DE-lncRNAs as strong functional candidates because their pronounced changes showed a statistically robust correlation with major phenotypic shifts in morphology and metabolite production. This approach provides a practical basis for generating mechanistic hypotheses. Therefore, the following discussion focuses on the high-priority candidates within this context. Among these candidates, lncRNA MSTRG.3124.1 was significantly more abundant in G01 (FPKM = 281.48) than in G02 (log_2_FC = −4.94; FDR < 1 × 10^−88^). We predicted that it drives the efficient synthesis of melanin and polysaccharides in G01 by regulating the expression of target genes such as glycosyltransferases. In contrast, several lncRNAs were specifically activated in G02, showing strong potential for negative regulation. Among them, MSTRG.10627.3 showed the most considerable differential expression (FDR < 1 × 10^−105^; log_2_FC = 10.53), and its high expression in G02 suggests that it is a key inhibitory switch in response to high salt stress conditions. MSTRG.4077.4 also exhibited very high differential expression (FPKM = 57.59; FDR < 1 × 10^−75^), and possibly acted as a primary inhibitory molecule in G02. MSTRG.4764.1 showed relatively high expression in G02 (FPKM = 65.61) and basal expression in both groups, and was possibly responsible for basal inhibition. These lncRNAs collectively reduced the metabolite content in G02 by inhibiting the expression of key enzyme genes involved in pathways such as glycolysis/gluconeogenesis (ko00010) and tyrosine metabolism (ko00350).

High melanin synthesis in G01 (3.22 g/L) strongly indicated the potent activation of the melanin synthesis pathway. Although the classical melanin synthesis pathway was not directly detected in the KEGG enrichment results, significant enrichment of target genes in the “peroxisome” pathway (ko04146) provided a critical clue. Peroxisomes are important intracellular oxidative metabolic sites that are involved in the degradation and synthesis of various substances, including melanin precursors [46,47]. Early enzymes involved in melanin biosynthesis (e.g., BcPKS12/13 and BcYGH1) are localized in peroxisomes [48]. Therefore, we hypothesized that specific lncRNAs in G01 create an oxidative microenvironment favorable for melanin synthesis, or directly regulate the expression of key melanin synthesis enzyme genes by cis-regulating the expression of peroxisome-related genes, thereby driving substantial melanin accumulation. Conversely, significant upregulation of lncRNAs, such as MSTRG.10627.3, and MSTRG.4077.4, under high salt stress conditions (G02) may inhibit these pathways, leading to decreased melanin production.

Analysis of the trans-acting target genes revealed another layer of a broad regulatory network. Significant enrichment in processes such as “DNA integration” and “protein ubiquitination” indicated that some lncRNAs were widely involved in maintaining genome stability and protein degradation processes via base complementary pairing. Protein ubiquitination is an important post-translational modification involved in various processes such as the cell cycle, signal transduction, and stress responses [49,50]. These results suggest that some highly expressed lncRNAs in G02 affect the activity of key salt stress response factors at the protein level via trans-acting mechanisms, thereby indirectly influencing morphogenesis and metabolism.

In conclusion, we constructed a preliminary regulatory model for lncRNAs involved in the salt stress response in *E. cristatum*. Under high-salt stress conditions, lncRNA expression profiles were reprogrammed. Some lncRNAs (e.g., MSTRG.3124.1) directly regulate adjacent metabolic genes related to sugar metabolism and melanin synthesis via cis-acting mechanisms, whereas others (e.g., MSTRG.10627.3 and MSTRG.4077.4) affect broad stress responses and protein degradation pathways via trans-acting mechanisms. The combined effects of these regulatory mechanisms may lead to the morphological transition of mycelia from the sexual to the asexual form and inhibit the synthesis of secondary metabolites such as polysaccharides and melanin.

## 5. Conclusions

Through integrated lncRNA sequencing and physiological phenotype analysis, this study systematically revealed the expression characteristics and potential regulatory functions of lncRNAs in *E. cristatum* under salt stress. A total of 203 lncRNAs were identified, 120 of which were closely associated with mycelial morphological differentiation and metabolite accumulation. The high expressed MSTRG.3124.1 in the high-yield G01 strain possibly promoted polysaccharide and melanin synthesis by regulating carbohydrate metabolism and the peroxisome pathway. In contrast, lncRNAs highly expressed in the G02 strain, such as MSTRG.10627.3 and MSTRG.4077.4, may negatively regulate secondary metabolism by inhibiting metabolic pathways and activating protein ubiquitination. These results suggest that lncRNAs play important roles in salt stress response and metabolic process regulation in *E. cristatum*, providing new insights into the environmental adaptation mechanisms of filamentous fungi and establishing a theoretical foundation for improving industrial strain performance via genetic modifications.

However, the proposed model requires further validation. Future work should not only seek to confirm the roles of the highly expressed candidate lncRNAs identified here but also explore the potential functions of lower-abundance DE-lncRNAs. In particular, the functions of key lncRNAs should be confirmed through gene knockout or overexpression experiments to observe their direct effects on target gene expression, mycelial morphology, and metabolite production. Furthermore, an integrated analysis of lncRNAs with other regulatory molecules, such as miRNAs and circular RNAs, would help construct a comprehensive competing endogenous RNA network, thereby systematically elucidating the molecular mechanisms governing the morphology and metabolism of *E. cristatum*.

## Figures and Tables

**Figure 1 biology-14-01592-f001:**
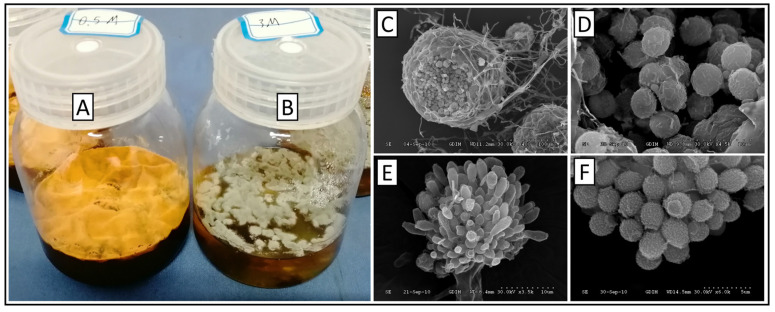
Fermentation broths and microscopic characteristics of the sexual and asexual morphs of *Eurotium cristatum*. (**A**) Sexual fermentation broth. (**B**) Asexual fermentation broth. (**C**) Cleistothecia. (**D**) Ascospores. (**E**) Conidial heads. (**F**) Conidia.

**Figure 2 biology-14-01592-f002:**
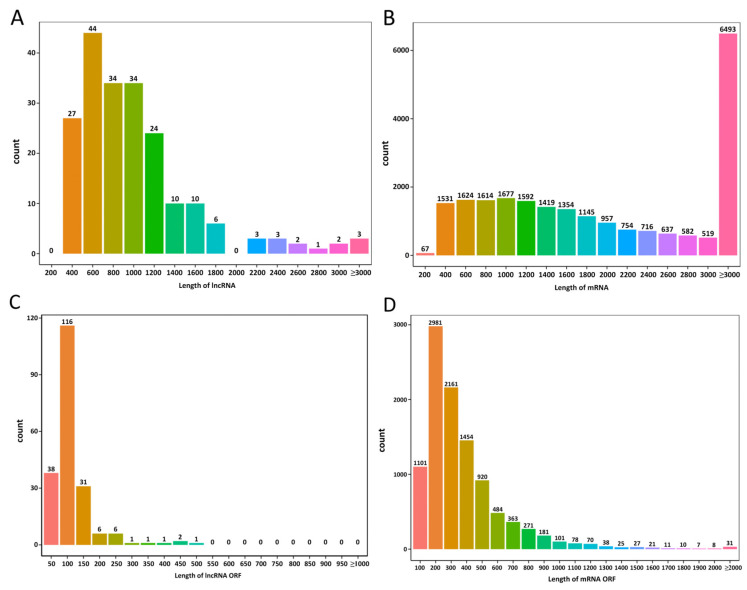
Characteristics of long non-coding (lncRNAs) and messenger RNAs (mRNAs) in *Eurotium cristatum*. (**A**) LncRNA length statistics. (**B**) mRNA length statistics. (**C**) LncRNA open reading frame (ORF) length statistics. (**D**) mRNA ORF length statistics. The horizontal axis represents length, while the vertical axis represents the number of RNAs within that length range.

**Figure 3 biology-14-01592-f003:**
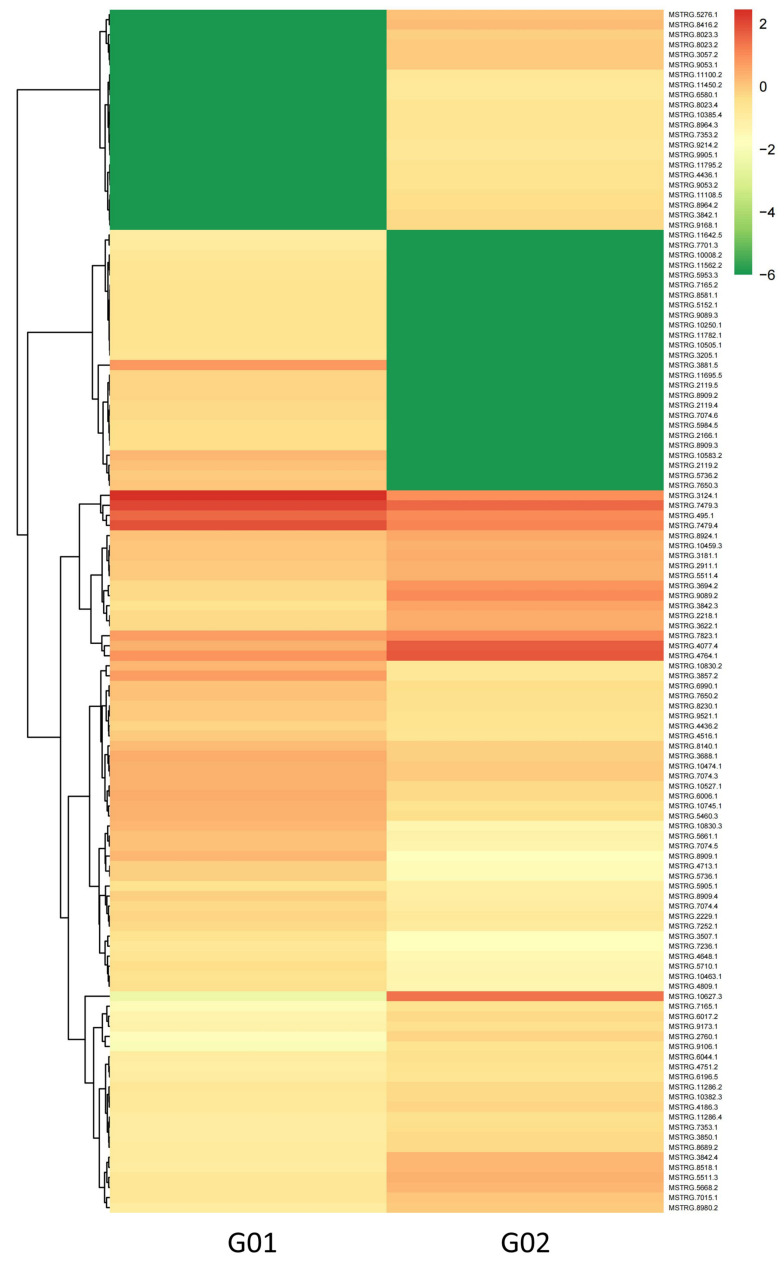
Clustering heatmap of differentially expressed long non-coding RNAs (DE-lncRNAs).

**Figure 4 biology-14-01592-f004:**
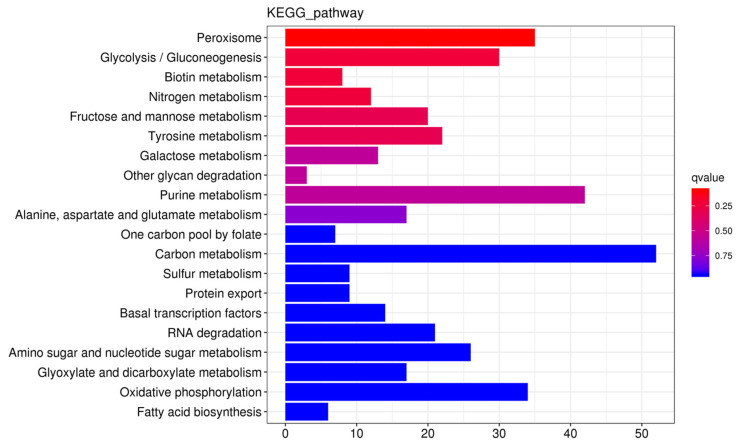
Kyoto Encyclopedia of Genes and Genomes (KEGG) enrichment bar plot of the cis-acting target genes of DE-lncRNAs. Note: The *x*-axis represents the gene number (number of genes of interest annotated to a specific term). The *y*-axis represents each pathway term. The bar color indicates the *p* value determined via the hypergeometric test.

**Figure 5 biology-14-01592-f005:**
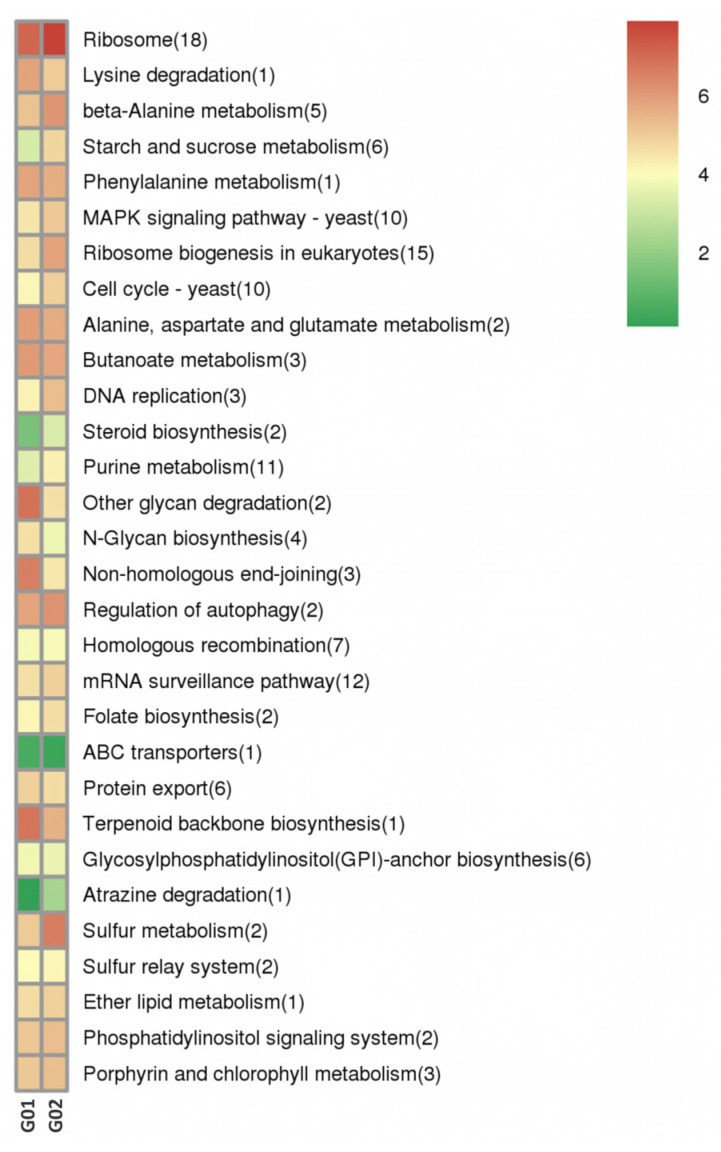
KEGG enrichment cluster plot of the trans-acting target genes of DE-lncRNAs.

**Figure 6 biology-14-01592-f006:**
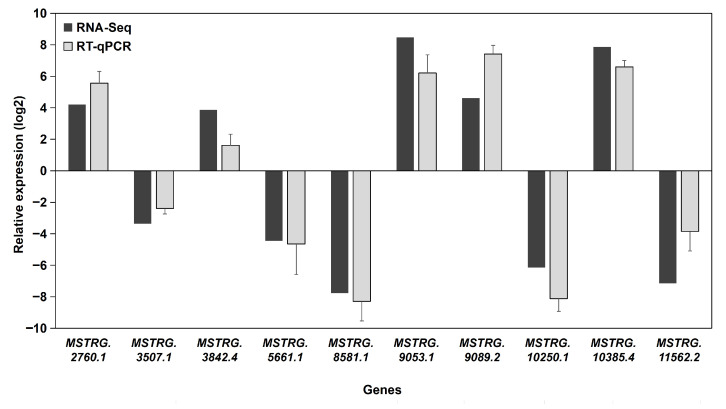
The relative expression of ten DE-lncRNAs was determined by RT-qPCR. Data are presented as the fold change in the asexual morph (G02) relative to the sexual morph (G01, which served as the calibrator), shown as mean ± SD (*n* = 3). The corresponding FPKM values from RNA-seq are presented for qualitative trend comparison. The concordant direction of change (up- or down-regulation) between the methodologies supports the reliability of the RNA-seq data.

**Figure 7 biology-14-01592-f007:**
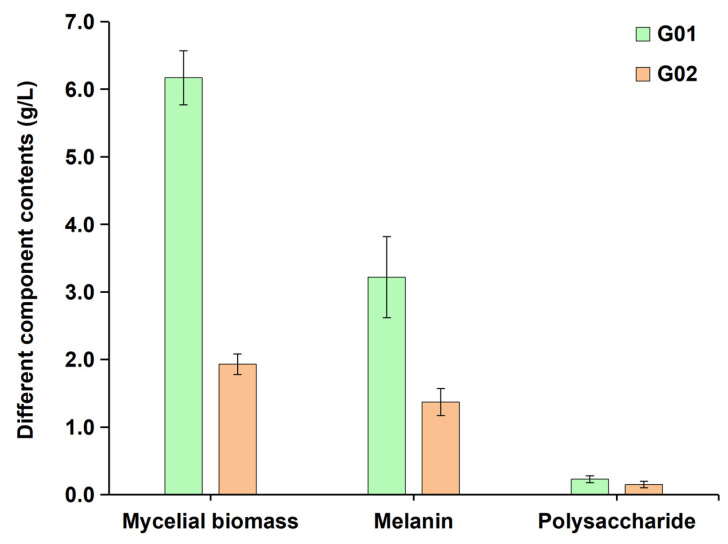
Contents of different components in the *Eurotium cristatum* fermentation broth.

**Table 1 biology-14-01592-t001:** Primer sequences of DE-lncRNAs used for RT-qPCR.

Gene ID	Type	Reverse Primer (3′-5′)	Reverse Primer (3′-5′)
MSTRG.2760.1	lincRNA	CTCACCACCTTCATCCAC	GCCCACCAAGTCCTATCA
MSTRG.3507.1	lincRNA	CGGAGAAAGTACCGATGT	GGTCAAAGCCGTGAGTGT
MSTRG.3842.4	antisense	TAGTTTAGCGTATCTTGGC	CACAGAGGTGGATTAGCA
MSTRG.5661.1	lincRNA	ACATACGCAAGTTAGTCGG	AAGATGGGAGTCTGGTTT
MSTRG.8581.1	antisense	TTGACGACTGGCACGATT	CGGAGTAGGAGTTTGGAGG
MSTRG.9053.1	lincRNA	TGAGCACCCGTTCAAGGC	AGGATCGGCATCAGGCAC
MSTRG.9089.2	lincRNA	CATTCGGAACGAGACACT	CTGCCTTGAATCATCCTG
MSTRG.10250.1	antisense	ACATACGGCATTCCAACA	CACAATTCAACGGATACAC
MSTRG.10385.4	lincRNA	ATAAATCCCGTGGTCCCT	ATACCCGTTGGTTCGTTG
MSTRG.11562.2	lincRNA	ACCCTGCCAATCTCCTAC	ATGTGCCTTGTATCTCCA

## Data Availability

The raw transcriptome sequencing data generated in this study have been deposited in the NCBI Sequence Read Archive (https://www.ncbi.nlm.nih.gov/sra/ accessed on 11 November 2025) under accession number PRJNA827193. All other supporting data are contained within the article and its Appendix A.

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
