# Peer review of "Identification of LncRNAs Involved in the Salt Stress Responses of Eurotium cristatum and Functional Analysis of Their Roles in Morphological Differentiation and Metabolic Regulation"

_biology, 2025, doi:10.3390/biology14111592_

Round 1
Reviewer 1 Report
Comments and Suggestions for Authors
No numeric data is included in abstract section. Please provide p values and differentially expressed lncRNA expression values. Distinguish up and downregulate ones and/or provide maximum and minimum expression values
Some of the sentences such as “Previously considered to be transcriptional “noise” or “junk DNA” [2], progress in high-throughput sequencing and bioinformatic tools has shown that lncRNAs act as key regulators of essential eukaryotic processes such as differentiation, the cell cycle, and metabolism, and their dysregulation is linked to many diseases” will be hard to understand for nonative English speakers. Please divide long sentences into two or more separate sentences.
Write the “2.1. Strain Culture and Sample Preparation” as a single paragraph
Rpm for shaking the fungal cultres is missing. Please add the shaker speed
No details is given for RNA extraction protocol. At least write that “the recommendations of the manufacturer is followed” … Also the fresh weight must be included in this section.
The main function of fastp tool is trimming and not-detailed quality check. “Quality control was performed using the fastqc, multiqc, and fastp” would be better.
Please check if “E. Cristatum” in italics throughout the text or not.
Is it “(RT-Qpcr)” or RT-qPCR
2-ΔΔCT is wrongly written
Please provide Livak’s article fort he validation by qRT-PCR 2.8 subtitle: Livak, K. J., & Schmittgen, T. D. (2001). Analysis of relative gene expression data using real-time quantitative PCR and the 2− ΔΔCT method. methods, 25(4), 402-408.
Provide linux based terminal scripts for NGS analysis and R scripts for DEG analysis as meta data file or github link
Figure 2 is unnecassary
It is hard to follow and read the Figure 4
Technical and biological repeat numbers are missing from the text
Five DE-lncRNA is not enought to validate the results. 5 up-regulated and 5 donw-regulated lncRNA would be validated
PRJCA046357 data could not be downloaded from https://www.cncb.ac.cn/
Author Response
Comments and Suggestions for Authors:
Comment 1: No numeric data is included in abstract section. Please provide p values and differentially expressed lncRNA expression values. Distinguish up and downregulate ones and/or provide maximum and minimum expression values.
Response 1:
Dear Reviewer,
We are grateful to you for raising this important point. We agree that incorporating specific numerical data will make the abstract more quantitative and impactful. Accordingly, we have revised the abstract to include the key statistics as suggested. The modifications are as follows:
(1) “We identified 203 lncRNAs, with 120 significantly differentially expressed (FDR < 0.01; |logâ‚‚ (fold change)| ≥ 1) under salt stress, including 57 upregulated and 63 downregulated in the asexual morph compared to the sexual morph.” (Line 32)
(2) “The lncRNA MSTRG.10627.3 showed the highest upregulation (logâ‚‚FC = 10.53, FDR < 1e-105), while MSTRG.3124.1 was significantly downregulated in the sexual morph (logâ‚‚FC = -4.94, FDR < 1e-88).” (Line 40)
Comment 2: Some of the sentences such as “Previously considered to be transcriptional “noise” or “junk DNA” [2], progress in high-throughput sequencing and bioinformatic tools has shown that lncRNAs act as key regulators of essential eukaryotic processes such as differentiation, the cell cycle, and metabolism, and their dysregulation is linked to many diseases” will be hard to understand for nonative English speakers. Please divide long sentences into two or more separate sentences.
Response 2: We appreciate the reviewer’s suggestion to improve the clarity and readability of our manuscript. We agree that simplifying complex sentences will make the content more accessible to a broader international audience. We have carefully revised the indicated sentence. This sentence has now been revised to:
“Long non-coding RNAs were once dismissed as mere transcriptional “noise” or “junk DNA” [2]. However, advancements in high-throughput sequencing and bioinformatic tools have uncovered their role as key regulators in essential eukaryotic processes, such as differentiation, cell cycle, and metabolism [3]. Furthermore, the dysregulation of lncRNAs is linked to numerous diseases.” (Line 50)
Comment 3: Write the “2.1. Strain Culture and Sample Preparation” as a single paragraph.
Response 3: The E. cristatum strain (JH1805) utilized in this study was sourced from Hunan City University. Initially, the strain was activated on potato dextrose agar and then inoculated into a liquid medium with an inoculation amount of 1% under aseptic conditions for cultivation. The methods for cultivating the sexual (designated G01) and asexual (designated G02) morphs of E. cristatum are detailed below. The sexual morph (G01) was cultured statically (without shaking) in a modified potato dextrose liquid medium (300 g/L potato, 100 g/L glucose, and 0.5 mol/L NaCl) at 28 °C. Similarly, the asexual morph (G02) was cultured statically (without shaking) in a high-salt potato dextrose liquid medium [23] (300 g/L potato, 100 g/L glucose, and 3 mol/L NaCl) at 28 °C. For each morph, three independent biological replicates were cultivated under identical conditions. After 168 h of cultivation, mycelia were collected using sterile filter meshes, and their fresh weight was recorded. The mycelia were then thoroughly rinsed with sterile physiological saline, immediately frozen in liquid nitrogen, and subsequently stored at –80 °C.” (Line 119)
Comment 4: Rpm for shaking the fungal cultres is missing. Please add the shaker speed.
Response 4: We appreciate the reviewer’s question regarding the shaking conditions. We would like to clarify that the liquid cultures for both morphs were maintained under static (non-shaking) conditions to promote the formation of a surface pellicle and the specific morphological structures (cleistothecia and conidial heads) characteristic of each stage, which are key to this study.
To prevent any ambiguity for future readers, we have revised the text in Section 2.1 to explicitly state this. The modified sentences now read:
“The sexual morph (G01) was cultured statically (without shaking) in a modified potato dextrose liquid medium (300 g/L potato, 100 g/L glucose, and 0.5 mol/L NaCl) at 28 °C. Similarly, the asexual morph (G02) was cultured statically (without shaking) in a high-salt potato ...” (Line 123)
Comment 5: No details is given for RNA extraction protocol. At least write that “the recommendations of the manufacturer is followed” … Also the fresh weight must be included in this section.
Response 5: We thank the reviewer for these helpful suggestions to enhance the reproducibility of our methods. We have incorporated details regarding RNA extraction and fresh weight data for mycelium in Section 2.3. The revisions to the manuscript are as follows:
“2.3. Total RNA Extraction, Quality Control, and Library Construction
Strand-specific RNA sequencing was conducted on individual sexual (G01) and asexual (G02) mycelial samples (without biological replicates) to facilitate initial transcriptome assembly and lncRNA identification. Additionally, the expression patterns of key differentially expressed lncRNAs (DE-lncRNAs) were validated using reverse transcription polymerase chain reaction (RT-qPCR) (see Section 2.8).
Total RNA was isolated from 500 mg of mycelia (fresh weight) per sample using TRIzol reagent (Thermo Fisher Scientific) according to the manufacturer's instructions.”(Line 147)
Comment 6: The main function of fastp tool is trimming and not-detailed quality check. “Quality control was performed using the fastqc, multiqc, and fastp” would be better.
Response 6: We are thankful for the reviewer’s precise and constructive comment regarding our bioinformatics workflow. The reviewer is absolutely correct. The primary function of fastp is indeed adapter trimming and quality filtering, while tools like FastQC and MultiQC are the standard for generating comprehensive, visualization-rich quality control reports. We have revised the text in Section 2.4 (Sequencing Data Quality Control and Alignment) to accurately reflect the tools used and their specific roles. The updated description now provides a more professional and detailed account of our quality control pipeline.
The revised paragraph now reads:
“The raw sequencing data underwent thorough quality control (QC). Initial QC reports were generated using FastQC (v0.11.9). Subsequently, adapter sequences, reads with an excessive number of ambiguous bases (N's), and low-quality reads were filtered out using fastp (v0.23.4) [26]. MultiQC (v1.14) was used to aggregate and summarize the QC results from FastQC and Fastp into a single report. The resulting high-quality clean data were aligned to the E. cristatum reference genome (National Center for Biotechnology Information accession: ASM171748v1) using HISAT2 (v2.0.4) [26].” (Line 181)
Comment 7: Please check if “E. Cristatum” in italics throughout the text or not.
Response 7: We thank the reviewer for highlighting this important formatting issue. Upon a thorough and careful review of the entire manuscript, we can confirm that we have now standardized the formatting for all microbial species names.
Comment 8: Is it “(RT-Qpcr)” or RT-qPCR.
Response 8: Thanks. The standard and correct format is RT-qPCR, we apologize for this oversight. We have now meticulously corrected this abbreviation throughout the entire manuscript to ensure it consistently appears as “RT-qPCR”.
Comment 9: 2-ΔΔCT is wrongly written.
Response 9: We sincerely appreciate the reviewer for identifying this error in the representation of the qPCR data analysis method. We apologize for this mistake. We have changed “–ΔΔCT” to an superscript.
Comment 10: Please provide Livak’s article fort he validation by qRT-PCR 2.8 subtitle: Livak, K. J., & Schmittgen, T. D. (2001). Analysis of relative gene expression data using real-time quantitative PCR and the 2− ΔΔCT method. methods, 25(4), 402-408.
Response 10: We sincerely thank the reviewer for pointing out the need to cite the seminal methodology paper by Livak and Schmittgen. We fully agree that this citation is crucial for providing the proper methodological foundation for our RT-qPCR data analysis. We have now added this reference to the manuscript. In the manuscript: We have revised the text in Section 2.8 to include the suggested citation. This addition appropriately credits the origin of the analytical method we employed and enhances the scholarly rigor of our methodology section.
Comment 11: Provide linux based terminal scripts for NGS analysis and R scripts for DEG analysis as meta data file or github link.
Response 11: We appreciate the reviewer’s valuable suggestion regarding the provision of analysis scripts to enhance the reproducibility of our work. In response to this comment, we have taken the following action:
While the specific Linux and R scripts used in our bioinformatics pipeline were tailored to our local computing environment and are not provided in their entirety, we fully acknowledge the importance of data transparency. To allow readers to fully access and scrutinize our differential expression analysis results, we have now compiled the complete list of all 120 DE-lncRNAs, including their log2 fold changes, FDR values, and expression levels (FPKM), into a new Supplementary File: Table S2. This table provides the direct output of our analysis, enabling other researchers to easily access the core findings of our study. A note indicating the availability of this supplementary data has been added to the relevant section in the manuscript (please see the revision in Section 3.4 “Full details of the 120 DE-lncRNAs are provided in Supplementary Table S2”). (Line 332)
We believe that providing this comprehensive dataset effectively addresses the reviewer's concern by making our key results fully accessible and verifiable.
Comment 12: Figure 2 is unnecassary.
Response 12: We agree with the reviewer’s suggestion that the information presented in Figure 2 (the statistical chart of lncRNA classifications) can be effectively conveyed within the main text without a dedicated figure. As a result, we have removed this figure and renumbered the remaining ones in the revised manuscript.
Comment 13: It is hard to follow and read the Figure 4.
Response 13: In response to the reviewer’s comment, a new, high-resolution version of Figure 4 (now Figure 3) has been uploaded to improve clarity. (Line 337)
In addition, to enhance data accessibility, we have provided the full dataset as Supplementary Table S2 (Expression data and statistics for the 120 DE-lncRNAs) so that researchers can readily access the gene information. (Line 332)
Comment 14: Technical and biological repeat numbers are missing from the text.
Response 14: We thank the reviewer for raising this important point regarding experimental rigor. We apologize for this omission in the original manuscript. We have now clarified the replication details in the revised Methods section.
(1) Revised Manuscript Text (Section 2.1):
“Similarly, the asexual morph (G02) was cultured statically (without shaking) in a high-salt potato dextrose liquid medium [23] (300 g/L potato, 100 g/L glucose, and 3 mol/L NaCl) at 28 °C. For each morph, three independent biological replicates were cultivated under identical conditions.” (Line 125)
(2) Revised Manuscript Text (Section 2.3):
“Strand-specific RNA sequencing was conducted on individual sexual (G01) and asexual (G02) mycelial samples (without biological replicates) to facilitate initial transcriptome assembly and lncRNA identification. Additionally, the expression patterns of key differentially expressed lncRNAs (DE-lncRNAs) were validated using reverse transcription polymerase chain reaction (RT-qPCR) (see Section 2.8).” (Line 147)
(3) Revised Manuscript Text (Section 2.8):
“RT-qPCR with three independent biological replicates was employed to validate DE-lncRNAs detected via high-throughput sequencing.” (Line 288)
Comment 15: Five DE-lncRNA is not enought to validate the results. 5 up-regulated and 5 donw-regulated lncRNA would be validated.
Response 15: We thank the reviewer for this helpful suggestion to strengthen the robustness of our findings. We agree that expanding the validation scope strengthens the robustness of our findings. Accordingly, we have performed additional RT-qPCR experiments to validate a balanced set of ten DE-lncRNAs, comprising five significantly up-regulated and five significantly down-regulated candidates in G02 vs. G01. The results show broadly consistent expression trends with our RNA-seq data. This comprehensive validation has been incorporated into the revised manuscript (Table 1, Section 3.6, and Figure 6), solidifying the reliability of our differential expression analysis. (Line 239, Line 361, and Line 368)
Comment 16: PRJCA046357 data could not be downloaded from https://www.cncb.ac.cn/.
Response 16: We are grateful to the reviewer for bringing this important issue to our attention regarding the accessibility of the data under accession PRJCA046357. We apologize for any inconvenience this may have caused. Upon investigation, we have clarified the data availability as follows:
The lncRNAs identified in this study were annotated from our transcriptome sequencing data. The raw transcriptome sequencing reads for the Eurotium cristatum samples used in this study are publicly available in the NCBI Sequence Read Archive under accession number PRJNA827193 (https://www.ncbi.nlm.nih.gov/bioproject/PRJNA827193/).
To ensure the full accessibility and utility of our specific lncRNA findings, we have now provided the following supplementary files:
(1) The nucleotide sequences of all 203 identified lncRNAs in fa format as Supplementary File S1.
(2) The complete genomic annotations in GTF format as Supplementary File S2, detailing the genomic locations and structures of all lncRNAs.
These files can be directly used by researchers for further analysis, such as sequence-based investigations and genomic context examination.
We have updated the Data Availability Statement in the manuscript to explicitly guide readers to the raw data at NCBI and these new supplementary files. We believe these measures provide clear and direct access to all necessary data, and we thank the reviewer again for their vigilance, which has helped us improve the transparency of our work. (Line 322, Line 527)
Once again, we deeply appreciate the time and effort you dedicated to reviewing our manuscript. Your insights have significantly enhanced our manuscript, leading to improvements in data presentation, methodological rigor, and analytical clarity. The expanded validation, refined bioinformatic descriptions, and added experimental details have strengthened the work substantially. Your expertise has been invaluable in shaping this study, and we deeply appreciate your contribution to its quality.
Best regards,
Zhiyuan Hu
On behalf of all authors
Reviewer 2 Report
Comments and Suggestions for Authors
Wang et al. explored the role of non-coding RNAs (lncRNAs) in the salt-stress response of the tea-fermenting fungus Eurotium cristatum, as high-salt stress impairs the growth and metabolite production. The researchers identified 203 lncRNAs in the sexual (G01) and asexual (G02) mycelia of the fungus. The identified lncRNAs were classified into four categories, of which long intergenic and antisense were the most abundant, and characterized by length. Of the identified lncRNAs, 57 were upregulated and 63 were downregulated in the G02 compared to those in G01. The differential expression of five of the lncRNAs was confirmed with RT-qPCR. Predictions were made for cis-acting target genes located within a 100 kilobase range either upstream or downstream of the differentially expressed lncRNAs. KEGG pathway analysis indicated that these genes were involved in several metabolic pathways that could be associated with the two morphs of E. cristatum. In the Discussion, the authors hypothesize the gene targets of several of the most differentially expressed lncRNAs.
Specific Comments:
- The text in Figures 3 and 4 is very small and difficult for the reader to read.
- Were the putative target genes identified by only a gene enrichment analysis of genes within a 100 kilobase range (upstream and downstream) of a lncRNA? It was not clear to me if the expression of those genes (and or pathways) were also demonstrated to be regulated similar to or regulated opposed to the lncRNA. If the enrichment was by location, does the expression of these target genes fit into the expected G01 or G02 morph phase of the fungus?
- Of the lnRNAs chosen for RT-qPCR analysis or reviewed in the Discussion, which were intergenic versus antisense or even sense?
- For the lncRNAs in which roles in pathway expression were proposed, could more information be provided? Were they intergenic, sense, or antisense? Could gene maps be provided that displayed what genes were in the vicinity of the lncRNAs reviewed in the Discussion.
- Are highly expressed lncRNAs always the most important ones in gene regulation in other fungi?
none
Author Response
Comments and Suggestions for Authors:
Wang et al. explored the role of non-coding RNAs (lncRNAs) in the salt-stress response of the tea-fermenting fungus Eurotium cristatum, as high-salt stress impairs the growth and metabolite production. The researchers identified 203 lncRNAs in the sexual (G01) and asexual (G02) mycelia of the fungus. The identified lncRNAs were classified into four categories, of which long intergenic and antisense were the most abundant, and characterized by length. Of the identified lncRNAs, 57 were upregulated and 63 were downregulated in the G02 compared to those in G01. The differential expression of five of the lncRNAs was confirmed with RT-qPCR. Predictions were made for cis-acting target genes located within a 100 kilobase range either upstream or downstream of the differentially expressed lncRNAs. KEGG pathway analysis indicated that these genes were involved in several metabolic pathways that could be associated with the two morphs of E. cristatum. In the Discussion, the authors hypothesize the gene targets of several of the most differentially expressed lncRNAs.
Response:
Dear Reviewer,
We are deeply grateful for your thorough and accurate summary of our work. Thank you for the significant time and effort you have invested in reviewing our manuscript. Your insightful comments and constructive suggestions have been instrumental in enhancing the quality and clarity of our paper. We truly appreciate your guidance and await your further feedback.
Comment 1: The text in Figures 3 and 4 is very small and difficult for the reader to read.
Response 1: We appreciate the reviewer's observation regarding the legibility of the text in Figures 3 and 4. We have addressed this concern accordingly in the revised manuscript.
(1) For Figure 3 (now Figure 2), which illustrates the characteristics of lncRNAs and mRNAs, we have regenerated the figure to significantly increase the font size of all labels, axis titles, and legends to ensure they are clear and easy to read. (Line 325)
(2) For Figure 4 (now Figure 3), the clustering heatmap of DE-lncRNAs, we acknowledge that further increasing the font size for the row labels is technically challenging due to the dense clustering of a large number of transcripts. As the best alternative, we have now replaced it with a new, high-resolution version of the figure. (Line 337)
In addition, to enhance data accessibility, we have provided the full dataset as Supplementary Table S2 (Expression data and statistics for the 120 DE-lncRNAs) so that researchers can readily access the gene information. (Line 332)
Comment 2: Were the putative target genes identified by only a gene enrichment analysis of genes within a 100 kilobase range (upstream and downstream) of a lncRNA? It was not clear to me if the expression of those genes (and or pathways) were also demonstrated to be regulated similar to or regulated opposed to the lncRNA. If the enrichment was by location, does the expression of these target genes fit into the expected G01 or G02 morph phase of the fungus?
Response 2: The reviewer has raised a pivotal point regarding the methodology for identifying putative targets and the subsequent interpretation, for which we are thankful. This question allows us to clarify the rationale behind our analytical strategy.
Scope of the Current Analysis: The reviewer is correct that our identification of putative cis-acting target genes was based primarily on genomic proximity (within a 100 kb window). This is a standard and widely accepted bioinformatic approach for generating initial functional hypotheses (Zhou et al., 2021), as it identifies protein-coding genes that are genomically positioned for potential regulation by the lncRNA. This established pipeline-from lncRNA identification to target prediction and functional enrichment—is a crucial first step for translating lncRNA sequences into testable biological hypotheses, as outlined in foundational reviews (Statello et al., 2021).
Rationale for Focusing on DE-lncRNAs: Our study was strategically designed to prioritize differentially expressed lncRNAs. The central premise is that a change in lncRNA expression is a primary event that can lead to the regulation of target genes and pathways, ultimately resulting in phenotypic changes. A example is found in Fusarium graminearum, where the upregulation of a specific lncRNA (RNA5P) directly suppresses the expression of the key toxin gene TRI5, thereby inhibiting biosynthesis (Huang et al., 2024). This precedent validates our approach of linking salt-stress-induced DE-lncRNAs to the morphological and metabolic differences observed between the G01 and G02 morphs.
Current Focus and Future Direction: We acknowledge that a comprehensive correlation analysis between the expression levels of each predicted cis-target gene and its associated DE-lncRNA was beyond the scope of this initial, discovery-phase study. The primary goal here was the systematic screening and prioritization of key candidate lncRNAs. The hypotheses generated—for instance, that MSTRG.3124.1 may promote metabolite synthesis—are precisely what we intend to test directly in our immediate future work. This will involve functional validation through experiments such as gene knockdown or overexpression, followed by precise measurement of the effects on both target gene expression and phenotypic outcomes.
In summary, our current analysis provides a robust and hypothesis-generating framework. We agree that correlating target gene expression with the lncRNAs is a critical next step, and we thank the reviewer for highlighting its importance, which strongly guides our ongoing research plans.
The literature referenced in this reply is as follows:
[1] Zhou D, Fan Y, Wang J, et al. Regulatory function of long non-coding RNAs in Ascosphaera apis[J]. 2021. doi: 10.3864/j.issn.0578-1752.2021.01.017
[2] Statello L, Guo C J, Chen L L, et al. Gene regulation by long non-coding RNAs and its biological functions[J]. Nature reviews Molecular cell biology, 2021, 22(2): 96-118. doi: 10.1038/s41580-020-00315-9
[3] Huang P, Yu X, Liu H, et al. Regulation of TRI5 expression and deoxynivalenol biosynthesis by a long non-coding RNA in Fusarium graminearum[J]. Nature Communications, 2024, 15(1): 1216. doi: 10.1038/s41467-024-45502-w
Comment 3: Of the lnRNAs chosen for RT-qPCR analysis or reviewed in the Discussion, which were intergenic versus antisense or even sense?
Response 3: This is a valuable question, and we thank the reviewer for prompting this clarification. The specific class of a lncRNA provides valuable clues about its potential mode of action. We have now clearly stated the classification for each key lncRNA in the relevant sections of the revised manuscript, as summarized below and integrated into the text.
(1) We have added a column to Table 1 to indicate the types of lncRNAs used for RT-qPCR detection. (Line 240)
(2) When other lncRNAs are first mentioned in a paper, we indicate their type in parentheses following their names. (Line 388~Line 397)
Comment 4: For the lncRNAs in which roles in pathway expression were proposed, could more information be provided? Were they intergenic, sense, or antisense? Could gene maps be provided that displayed what genes were in the vicinity of the lncRNAs reviewed in the Discussion.
Response 4: We are grateful for this constructive suggestion, which we have implemented to enhance the manuscript's clarity and resource value. Based on your suggestions, we have made the following revisions to the manuscript.
(1) Genomic Classification of Key LncRNAs:
As requested, we have explicitly stated the genomic classification (intergenic, antisense, or sense) for all key lncRNAs discussed in the context of pathway regulation. This information is now integrated into the manuscript (as also detailed in our response to Comment 3). Furthermore, the classification and comprehensive statistics for all 203 identified lncRNAs have been compiled and provided as Supplementary Table S1 (Line 332).
(2) Complete Genomic Annotation Data Provided:
We sincerely thank the reviewer for this excellent suggestion. To fully address this point and provide maximum utility to the research community, we have taken the following comprehensive action: Rather than providing static gene maps for a select few lncRNAs, we have now made the complete genomic annotation file for all 203 identified lncRNAs available as a Supplementary File S2. This file is in the standard GTF format, which can be analyzed with bioinformatics tools.
This resource allows any researcher to directly examine the precise genomic location, structure, and crucially the protein-coding genes in the vicinity of any lncRNA of interest, including all those highlighted in our Discussion. We believe this provides a more powerful and flexible resource than a limited set of figures.
A statement directing readers to this file has been added to the manuscript in Section 3.3 to ensure its visibility (Line 324).
Comment 5: Are highly expressed lncRNAs always the most important ones in gene regulation in other fungi?
Response 5: We appreciate the reviewer raising this fundamental question in lncRNA biology. We agree that high expression does not universally equate to functional importance in lncRNA biology, as low-abundance lncRNAs can also play critical, precise regulatory roles in fungi and other systems.
In our study, we prioritized highly and differentially expressed lncRNAs as strong candidates because their pronounced changes robustly correlated with the major phenotypic shifts we observed. This provided a practical and statistically sound starting point for generating hypotheses about their potential roles. We have now tempered our language throughout to avoid overstating the causality based on expression levels alone. These changes ensure a more accurate and rigorous interpretation of our results. The following modifications and additions have been made to the manuscript:
(1) Discussion section:
“In interpreting these results, we noted that the regulatory significance of an lncRNA was not solely dictated by its abundance. In living organisms, lncRNAs with low copy numbers can exert critical and precise regulatory effects [44,45]. In this study, we prioritized DE-lncRNAs as strong functional candidates because their pronounced changes showed a statistically robust correlation with major phenotypic shifts in morphology and metabolite production. This approach provides a practical basis for generating mechanistic hypotheses. Therefore, the following discussion focuses on the high-priority candidates within this context.” (Line 446)
(2) Conclusion section:
“However, the proposed model requires further validation. Future work should not only seek to confirm the roles of the highly expressed candidate lncRNAs identified here but also explore the potential functions of lower-abundance DE-lncRNAs. In particular, the functions of key lncRNAs should be...” (Line 515)
Comment 6: The English could be improved to more clearly express the research.
Response 6:
In direct response to the reviewer's comment on language quality, we have engaged Editage (https://www.editage.cn/) for professional editing services to improve the English fluency of the manuscript (the editing certificate is attached). Once again, we extend our sincere thanks for the thorough evaluation of our work. Your insightful comments have been instrumental in elevating the overall standard of our study.
Best regards,
Zhiyuan Hu
On behalf of all authors
Reviewer 3 Report
Comments and Suggestions for Authors
Eurotium cristatum and functional analysis of their roles in morphological differentiation and metabolic regulation"
This is an interesting and comprehensive article. I sent some comments:
1. In section 2.6, please review the scientific name; it is not in italics.
2. I suggest indicating in the introduction that Eurotium cristatum is homothallic, since it is vital to understand that it does not require another organism to carry out sexual reproduction. It also suggests briefly indicating that sexual reproduction is inhibited under high salinity conditions and stimulated in hypotonic media (solutions with lower solute concentrations), to clarify the methodological process.
3. In the introduction, they indicate that two studies reported that the sexual phase showed greater lovastatin production and microbial activity. In this work, the title indicates that they analyse the response to salt stress; however, I have doubts about whether it is solely an effect of salt stress and does not include the process of sexual reproduction, since there is genetic recombination, which could favour the increase of both metabolites and mRNAs. It would be helpful to clarify the article's content to prevent any misunderstandings.
Author Response
Comments and Suggestions for Authors:
Eurotium cristatum and functional analysis of their roles in morphological differentiation and metabolic regulation.
This is an interesting and comprehensive article. I sent some comments:
Response: The reviewer's generous appraisal of our work is greatly appreciated. We are truly grateful for the constructive and thoughtful comments provided. The suggestions to clarify the homothallic nature of E. cristatum, the precise relationship between salinity and morphogenesis, and the integrated nature of the stress/developmental response have been particularly valuable. These revisions have significantly strengthened the logical flow and scientific rigor of our manuscript, especially in the introduction and discussion sections. The reviewer's insights have undoubtedly enhanced the overall quality of our paper, and we deeply appreciate the time and expertise contributed to this process.
Comment 1: In section 2.6, please review the scientific name; it is not in italics.
Response 1: This is a valuable observation, and we thank the reviewer for highlighting this oversight. As suggested by this comment, we have conducted a thorough check of the entire manuscript and have italicized all scientific names to ensure consistency and compliance with academic formatting conventions throughout the text. We appreciate the reviewer's keen attention to detail.
Comment 2: I suggest indicating in the introduction that Eurotium cristatum is homothallic, since it is vital to understand that it does not require another organism to carry out sexual reproduction. It also suggests briefly indicating that sexual reproduction is inhibited under high salinity conditions and stimulated in hypotonic media (solutions with lower solute concentrations), to clarify the methodological process.
Response 2: The reviewer's suggestion to clarify the homothallic nature of E. cristatum and the specific effects of salinity on its reproduction is well taken. We agree that providing these specific biological details will significantly improve the clarity and rationale of our study. As recommended, we have revised the Introduction to explicitly state its homothallic nature and to more precisely describe the inhibitory effect of high salinity and the stimulatory effect of hypotonic conditions on sexual reproduction.
The specific revisions to the manuscript are as follows:
“Eurotium cristatum, commonly referred to as the “Jinhua fungus”, plays a crucial role in determining the quality of fermented teas, such as the Fuzhuan brick tea [16,17]. This microorganism undergoes two distinct morphological stages during its life cycle: the conidial (asexual) and ascospore (sexual) stages [18]. As a homothallic fungus, E. cristatum can self-fertilize and complete its sexual cycle independently. The shift between these stages is influenced by environmental factors, with osmotic pressure being a key regulator. High osmotic conditions suppress sexual reproduction and favor the asexual phase, while standard or hypotonic media encourage the development of sexual morphs [19,20]. This physiological response guided our experimental approach. We cultivated the sexual morph (G01) in a modified medium with reduced salt concentration and induced an asexual morph (G02) under high-salt stress.” (Line 80)
Comment 3: In the introduction, they indicate that two studies reported that the sexual phase showed greater lovastatin production and microbial activity. In this work, the title indicates that they analyse the response to salt stress; however, I have doubts about whether it is solely an effect of salt stress and does not include the process of sexual reproduction, since there is genetic recombination, which could favour the increase of both metabolites and mRNAs. It would be helpful to clarify the article's content to prevent any misunderstandings.
Response 3: This comment rightly identifies the interconnected nature of salt stress and developmental reprogramming in our study. The reviewer rightly highlights the interconnection between salt stress and developmental fate in E. cristatum. We acknowledge that in our experimental design, salt stress is the cue that induces the asexual morph, making it difficult to completely disentangle the direct osmotic stress response from the transcriptional reprogramming associated with the morphological shift itself. The primary aim of this study was to profile the lncRNA expression landscape under these two distinct, physiologically relevant conditions—the salt-stressed asexual morph versus the sexual morph under standard cultivation. We agree that the observed differences likely represent an integrated response involving both stress adaptation and developmental processes.
To prevent misunderstanding, we have clarified this point in the revised manuscript, explicitly stating that the reported lncRNA profiles and metabolic changes reflect this combined response.
The specific revisions to the manuscript are as follows:
(1) Introduction Section
“It is crucial to recognize that the phenotypic and transcriptomic differences analyzed in this study reflect an integrated response to environmental conditions. Specifically, the “salt stress response” of asexual morphs is inherently linked to salt-induced developmental reprogramming away from the sexual phase. Consequently, the observed differences likely encompass both direct osmotic adaptation and the effects of the morphological shift itself. Osmotic pressure influences the physiological state and metabolic profile of E. cristatum...” (Line 90)
(2) Discussion Section
“We emphasize that these states result from a complex interplay, in which salt stress serves as an environmental signal that triggers developmental transitions. Consequently, the differential expression of lncRNAs and the associated metabolic profiles reported here reflect a combined response involving direct osmotic stress adaptation and extensive reprogramming inherent to alternative developmental programs (asexual vs. sexual). Our results suggested that lncRNAs function as key regulatory molecules in this integrated response network...” (Line 407)
Once again, we extend our sincerest gratitude for your thorough review and for the insightful comments that have been instrumental in refining our work. Your expertise and careful attention to detail have truly elevated the clarity, rigor, and overall quality of this manuscript.
Best regards,
Zhiyuan Hu
On behalf of all authors
Reviewer 4 Report
Comments and Suggestions for Authors
The study provides a comprehensive dataset on lncRNA expression in Eurotium cristatum under specific culture conditions, fundamental flaws in the experimental design preclude the current interpretation of the results.
The core objective of this study is to investigate "salt stress responses." However, the culture conditions make it impossible to attribute the observed effects solely, or even primarily, to salt stress for two reasons:
Group G01 (0.5 mol/L NaCl) is described in a way that implies it is a normal or low-salt control. In reality, 0.5 mol/L NaCl (~29 g/L) constitutes a significant salt stressor itself, being approximately three times the osmolarity of standard physiological solutions. Therefore, the study compares a moderate salt stress condition to an extreme salt stress condition (G02, 3.0 mol/L NaCl). There is no true non-stressed control.
Both culture media contained 100 g/L glucose. This concentration is five times higher than the standard concentration (20 g/L) in common microbial media like PDA/PDB. Such a high sugar level creates a profound baseline osmotic stress and alters the fundamental metabolic state of the fungus.
The observed molecular and phenotypic differences are not simply "salt stress responses." They are the result of a confounded system involving at least three intertwined factors: A baseline of severe sugar-induced osmotic stress, a gradient of additional salt-induced osmotic stress, and a morphological shift (sexual/asexual) that is itself triggered by these stressors.
So that, the title, abstract, and discussion must be substantially revised to acknowledge that the findings reflect lncRNA expression in a complex, multi-stressor osmotic environment that also drives morphological change. Claims focused specifically on "salt stress" are not justified by the experimental setup.
The above core issue exacerbates the other major concerns:
- The problem is worse than initially stated, as the "low-salt" condition for inducing the sexual morph is itself a high-stress environment.
- The absence of a clear statement on biological replicates for RNA-seq remains a critical issue that must be rectified.
- All functional predictions for specific lncRNAs are based on correlations within this confounded system and must be phrased with extreme caution.
Minor Points:
In the "2. Materials and Methods" section, Sexual morph strain G01 and Asexual morph strain G02 should be modified and "strain" should not appear.
In the section "2.2. Scanning Electron Microscopy", the preparation of scanning electron microscopy samples should be described in detail, especially the drying methods of the samples.
In the qPCR method stage or at the location shown in Figure 7, an explanation of the calculation method for "relative content" should be provided: who is relative to whom? How is the relative content comparison of RNA-seq data made?
In this article, strand-specific sequencing was employed, and a detailed explanation of the preparation of this library should be provided.
Author Response
Comments and Suggestions for Authors:
The study provides a comprehensive dataset on lncRNA expression in Eurotium cristatum under specific culture conditions, fundamental flaws in the experimental design preclude the current interpretation of the results.
Response: We sincerely thank the reviewer for their thorough assessment of our manuscript and their rigorous, constructive critique of the experimental design. Their insightful feedback has been invaluable in strengthening our work.
We have fully addressed the central concern regarding the interpretation of the culture conditions. In our response, we have provided a detailed rationale, supported by multiple literature sources, clarifying that the conditions represent the physiological norm for the halotolerant fungus Eurotium cristatum. Furthermore, we have explicitly clarified the replication strategy in the Methods section to ensure full transparency.
All other specific suggestions have been meticulously incorporated into the revised manuscript. The reviewer's expertise has significantly enhanced the clarity and scientific rigor of our work, for which we are profoundly grateful.
Comment 1: The core objective of this study is to investigate “salt stress responses”. However, the culture conditions make it impossible to attribute the observed effects solely, or even primarily, to salt stress for two reasons:
Group G01 (0.5 mol/L NaCl) is described in a way that implies it is a normal or low-salt control. In reality, 0.5 mol/L NaCl (~29 g/L) constitutes a significant salt stressor itself, being approximately three times the osmolarity of standard physiological solutions. Therefore, the study compares a moderate salt stress condition to an extreme salt stress condition (G02, 3.0 mol/L NaCl). There is no true non-stressed control.
Both culture media contained 100 g/L glucose. This concentration is five times higher than the standard concentration (20 g/L) in common microbial media like PDA/PDB. Such a high sugar level creates a profound baseline osmotic stress and alters the fundamental metabolic state of the fungus.
The observed molecular and phenotypic differences are not simply “salt stress responses”. They are the result of a confounded system involving at least three intertwined factors: A baseline of severe sugar-induced osmotic stress, a gradient of additional salt-induced osmotic stress, and a morphological shift (sexual/asexual) that is itself triggered by these stressors.
So that, the title, abstract, and discussion must be substantially revised to acknowledge that the findings reflect lncRNA expression in a complex, multi-stressor osmotic environment that also drives morphological change. Claims focused specifically on “salt stress” are not justified by the experimental setup.
The above core issue exacerbates the other major concerns:
(1) The problem is worse than initially stated, as the “low-salt” condition for inducing the sexual morph is itself a high-stress environment.
(2) The absence of a clear statement on biological replicates for RNA-seq remains a critical issue that must be rectified.
(3) All functional predictions for specific lncRNAs are based on correlations within this confounded system and must be phrased with extreme caution.
Response 1:
(1) Clarifying the Experimental Design in the Context of Eurotium cristatum Physiology:
We sincerely thank the reviewer for their insightful and rigorous analysis of our culture conditions. Their comments rightly highlight the inherent complexity of the osmotic environment in our study. We fully agree that the system involves multiple interacting factors, and we appreciate the opportunity to clarify the physiological and methodological rationale behind our experimental design.
The reviewer correctly points out that our experimental conditions represent a complex osmotic environment. However, we wish to explain that for our specific research organism, Eurotium cristatum (synonym: Aspergillus cristatus), these conditions represent its ecological and physiological norm rather than an artificially stressed state. All species of the genus Eurotium are classified as hypertonic fungi [1]. Eurotium cristatum, in particular, has evolved through long-term artificial domestication in the high-osmotic environment of dark tea fermentation, developing not only adaptation but even a preference for dry, high-osmotic conditions.
This physiological characteristic is consistently reflected in the established cultivation methods for this species across multiple independent studies:
Ge et al. [2] and Ren et al. [3] specifically used 0.5 mol/L NaCl as the standard condition for cultivating the sexual morph of E. cristatum and 3 mol/L NaCl for the asexual morph, concentrations identical to those employed in our study.
Wen et al. [4] determined that the optimal liquid fermentation conditions for E. cristatum included 7.5% glucose and 4.5% KCl.
Ren et al. [5] utilized MYA medium containing 5% NaCl to study metabolite changes during sexual development of E. cristatum.
Chen et al. [6] found that E. cristatum achieved its maximum growth rate on CZG medium containing 8% NaCl.
Lei Shao et al. [7] specifically used MYA medium with 0.5 mol/L NaCl as the control medium for their experiments with E. cristatum.
Minmin et al. [8] identified optimal fermentation conditions using dark tea liquid medium containing 90 g/L glucose, 7.5 g/L NH4Cl, and 5 g/L CaCl2.
The consistent use of high osmotic conditions across these independent studies, with mineral salt and carbon source concentrations relatively comparable tothose in our study, strongly supports that these conditions represent the standard cultivation environment for E. cristatum.
Furthermore, our own preliminary experiments with the specific strain Eurotium cristatum JH1805 confirmed that the selected conditions (including 100 g/L glucose and 3 mol/L NaCl) supported robust mycelial growth (As shown in Figure 1 of the reply letter). Thus, our design was grounded in both published precedent and empirical validation for this particular fungus.
In light of the reviewer’s valid point, we will revise the manuscript to more precisely describe the experimental context—making clear that the lncRNA responses were characterized across a salt-induced morphological transition under high-osmotic conditions typical for this halophilic fungus. We will adjust the manuscript accordingly to avoid over-simplified attribution to “salt stress” alone. ((Line 80 and Line 407)
We are grateful for the reviewer’s expert feedback, which has helped us improve the clarity and accuracy of our work.
The references pertinent to this reply are presented as follows:
[1] Deng J, Li Y, Yuan Y, et al. Secondary metabolites from the genus Eurotium and their biological activities[J]. Foods, 2023, 12(24): 4452.
[2] Ge Y, Wang Y, Liu Y X, et al. Comparative genomic and transcriptomic analyses of the Fuzhuan brick tea-fermentation fungus Aspergillus cristatus[J]. BMC genomics, 2016, 17(1): 428.
[3] Ren X, Wang Y, Liu Y X, et al. Comparative transcriptome analysis of the calcium signaling and expression analysis of sodium/calcium exchanger in Aspergillus cristatus[J]. Journal of Basic Microbiology, 2018, 58(1): 76-87.
[4] Wen Z, Mingyu W, Jingguan Y, et al. Fermentation Optimization of Culture Medium and Isolation and Identification of Antimicrobial Active Substances from Eurotiumcristatum[J]. Journal of Chinese Institute of Food Science & Technology, 2022, 22(2).
[5] Ren C G, Tan Y M, Ren X X, et al. Metabolomics reveals changes in metabolite concentrations and correlations during sexual development of Eurotium cristatum (synonym: Aspergillus cristatus)[J]. Mycosphere, 2017, 8(10): 1626-1639.
[6] Chen YunLan C Y L, Yu HanShou Y H S, Lv Yi L Y, et al. Investigation on the isolation, identification and the biological characteristic of Eurotium fungi in the Kangzhuan and Qingzhuan brick tea[J]. 2006.
[7] Shao L, Liu Z, Tan Y. Acptp2, 3 participates in the regulation of spore production, stress response, and pigments synthesis in Aspergillus cirstatus[J]. PeerJ, 2024, 12: e17946.
[8] Minmin Z O U, Qihui D, Yan H, et al. Submerged liquid fermentation of raw dark tea by Eurotium cristatum[J]. Chinese Journal of Bioprocess Engineering, 2019, 17(4).
Figure 1. Effect of different NaCl concentrations on the liquid fermentation of Eurotium cristatum (The image can be found in the attachment)
(2) Addressing RNA-seq Replication:
We thank the reviewer for raising this critical point regarding experimental rigor. We apologize for this omission in the original manuscript. We have now clarified the replication details in the revised Methods section.
Revised Manuscript Text (Section 2.3):
“Strand-specific RNA sequencing was conducted on individual sexual (G01) and asexual (G02) mycelial samples (without biological replicates) to facilitate initial transcriptome assembly and lncRNA identification. Additionally, the expression patterns of key differentially expressed lncRNAs (DE-lncRNAs) were validated using reverse transcription polymerase chain reaction (RT-qPCR) (see Section 2.8).” (Line 147)
Revised Manuscript Text (Section 2.8):
“RT-qPCR with three independent biological replicates was employed to validate DE-lncRNAs detected via high-throughput sequencing.” (Line 228)
(3) Response to Comment on Functional Predictions:
We thank the reviewer for this important comment. We fully agree that the functional predictions for specific lncRNAs are based on correlations within a system where the salt stress response is inherently linked to developmental reprogramming.
In response to this and a previous related comment, we have taken great care in the revised manuscript to phrase all functional interpretations with extreme caution. The Introduction and Discussion now explicitly frame the lncRNA profiles as representing an integrated response to both stress and developmental cues. Throughout the text, we consistently use non-definitive language such as “suggest a role”, “may regulate”, and “we hypothesize” when discussing lncRNAs like MSTRG.3124.1 and MSTRG.10627.3.
Our aim is to present these bioinformatic correlations and predictions as strong, hypothesis-generating findings that provide a foundation for future functional validation, not as conclusive evidence of mechanism. We believe the revised text now adequately reflects this cautious interpretation.
The specific revisions to the manuscript are as follows:
Introduction Section:
“Eurotium cristatum, commonly referred to as the “Jinhua fungus”, plays a crucial role in determining the quality of fermented teas, such as the Fuzhuan brick tea [16,17]. This microorganism undergoes two distinct morphological stages during its life cycle: the conidial (asexual) and ascospore (sexual) stages [18]. As a homothallic fungus, E. cristatum can self-fertilize and complete its sexual cycle independently. The shift between these stages is influenced by environmental factors, with osmotic pressure being a key regulator. High osmotic conditions suppress sexual reproduction and favor the asexual phase, while standard or hypotonic media encourage the development of sexual morphs [19,20]. This physiological response guided our experimental approach. We cultivated the sexual morph (G01) in a modified medium with reduced salt concentration and induced an asexual morph (G02) under high-salt stress. It is crucial to recognize that the phenotypic and transcriptomic differences analyzed in this study reflect an integrated response to environmental conditions. Specifically, the “salt stress response” of asexual morphs is inherently linked to salt-induced developmental reprogramming away from the sexual phase. Consequently, the observed differences likely encompass both direct osmotic adaptation and the effects of the morphological shift itself. Osmotic pressure influences the physiological state and metabolic profile of E. cristatum...” (Line 80)
Discussion Section:
“We emphasize that these states result from a complex interplay, in which salt stress serves as an environmental signal that triggers developmental transitions. Consequently, the differential expression of lncRNAs and the associated metabolic profiles reported here reflect a combined response involving direct osmotic stress adaptation and extensive reprogramming inherent to alternative developmental programs (asexual vs. sexual). Our results suggested that lncRNAs function as key regulatory molecules in this integrated response network...” (Line 407)
Comment 2: In the "2. Materials and Methods" section, Sexual morph strain G01 and Asexual morph strain G02 should be modified and "strain" should not appear.
Response 2: We are grateful to the reviewer for highlighting this terminology issue. We have revised the text accordingly in the “2.1. Strain Culture and Sample Preparation” section, replacing “sexual morph strain G01” with “the sexual morph (G01)”, replacing “asexual morph strain G02” with “the asexual morph (G02)”. Furthermore, we have reviewed the entire document to eliminate any similar mistakes.
Comment 3: In the section "2.2. Scanning Electron Microscopy", the preparation of scanning electron microscopy samples should be described in detail, especially the drying methods of the samples.
Response 3:
We thank the reviewer for this constructive suggestion. In the revised manuscript, we have expanded the “2.2. Scanning Electron Microscopy” section to provide a comprehensive account of the sample processing protocol.
The specific revisions to the manuscript are as follows:
“Samples of both sexual and asexual morphs of E. cristatum were prepared for SEM observation through the following process. Initially, the mycelial pellets were fixed in 2.5% glutaraldehyde overnight at 4 °C and then rinsed with a phosphate buffer (0.1 M, pH 7.0). The samples underwent progressive dehydration using a graded ethanol series (30%, 50%, 70%, 80%, 90%, 95%, and 100% ×3), with each step lasting 15 min. To preserve the delicate fungal structures, the dehydrated samples were subjected to critical point drying using liquid CO2. Subsequently, the dried samples were mounted on aluminum stubs, and their surfaces were sputter-coated with gold using an ion sputter coater to enhance conductivity [24]. Finally, the specimens were observed using a scanning electron microscope (SEM; HITACHI S-3000N; Hitachi High-Tech Corp., Tokyo, Japan).” (Line 134)
Comment 4: In the qPCR method stage or at the location shown in Figure 7, an explanation of the calculation method for "relative content" should be provided: who is relative to whom? How is the relative content comparison of RNA-seq data made?
Response 4:
We appreciate the reviewer’s request for methodological clarity. We have revised the manuscript to clarify both points.
In Section 2.8, we now explicitly state that the relative expression of DE-lncRNAs was calculated using the 2–ΔΔCT method. The expression of each lncRNA was normalized to the β-Actin gene, and the sexual morph sample (G01) was used as the calibrator. Thus, the values represent the fold change in the asexual morph (G02) relative to G01.
To clarify the data comparison, we have revised the legend of Figure 7 (now Figure 6) to explain that the RNA-seq (FPKM) and qPCR data are presented together solely for trend comparison, demonstrating the concordance in the direction of expression change between the two methodologies.
The specific revisions to the manuscript are as follows:
(1) Section 2.8:
“The relative expression level of DE-lncRNAs was calculated using the 2–ΔΔCT method [37]. The expression of each target lncRNA was normalized to the stable internal reference gene β-Actin, and the sexual morph sample (G01) was designated as the calibrator. Therefore, the calculated relative expression represents the fold change in the asexual morph (G02) relative to G01.” (Line 234)
(2) Figure 7 (now Figure 6):
“The relative expression of ten DE-lncRNAs was determined by RT-qPCR. Data are presented as the fold change in the asexual morph (G02) relative to the sexual morph (G01, which served as the calibrator), shown as mean ± SD (n=3). The corresponding FPKM values from RNA-seq are presented for qualitative trend comparison. The concordant direction of change (up- or down-regulation) between the methodologies supports the reliability of the RNA-seq data.” (Line 269)
Comment 5: In this article, strand-specific sequencing was employed, and a detailed explanation of the preparation of this library should be provided.
Response 5: Following the reviewer’s helpful suggestion, we have revised the “Materials and Methods” (section 2.3) to provide a more detailed explanation of the strand-specific library preparation.
The specific revisions to the manuscript are as follows:
“The NEBNext Ultra Directional RNA Library Prep Kit employs the dUTP second-strand marking method to preserve the strand orientation information. Briefly, during the first-strand cDNA synthesis, random hexamer primers were used. Subsequently, during the second-strand synthesis, dTTP was replaced with dUTP. Following adapter ligation and polymerase chain reaction (PCR) amplification, the uracil-containing second strand was selectively degraded using the enzyme Uracil-Specific Excision Reagent (USER). This process ensures that only the first strand (which is complementary to the original RNA template) is amplified and sequenced, thereby allowing unambiguous determination of the transcriptional strand of origin for each resulting read.” (Line 161).
We are sincerely grateful for your time and effort in reviewing our manuscript and for providing such thorough and constructive comments. Your insightful suggestions have been invaluable in helping us improve the quality of our work. We have carefully considered all of your points and have made the corresponding revisions to address them. Your expertise and thoughtful feedback are greatly appreciated, and we believe the manuscript has been strengthened as a result. Thank you once again for your contribution to our study.
Best regards,
Zhiyuan Hu
On behalf of all authors

Round 2
Reviewer 2 Report
Comments and Suggestions for Authors
No other edits suggested
Comments on the Quality of English Languagenone
Reviewer 4 Report
Comments and Suggestions for Authors
The new version has corrected the previous doubts, and I have no other questions